# Long-Term Aging of Concentrated Aqueous Graphene Oxide Suspensions Seen by Rheology and Raman Spectroscopy

**DOI:** 10.3390/nano12060916

**Published:** 2022-03-10

**Authors:** Benjámin Gyarmati, Shereen Farah, Attila Farkas, György Sáfrán, Loredana Mirela Voelker-Pop, Krisztina László

**Affiliations:** 1Department of Physical Chemistry and Materials Science, Faculty of Chemical Technology and Biotechnology, Budapest University of Technology and Economics, Műegyetem rkp. 3, H-1111 Budapest, Hungary; gyarmati.benjamin@vbk.bme.hu (B.G.); sfarah@edu.bme.hu (S.F.); 2Department of Organic Chemistry and Technology, Faculty of Chemical Technology and Biotechnology, Budapest University of Technology and Economics, Műegyetem rkp. 3, H-1111 Budapest, Hungary; farkas.attila@vbk.bme.hu; 3Research Institute for Technical Physics and Materials Science, Eötvös Loránd Research Network, Konkoly Thege M. út 29-33, H-1121 Budapest, Hungary; safran.gyorgy@energia.mta.hu; 4Anton Paar Germany GmbH, Helmuth-Hirth-Strasse 6, D-73760 Ostfildern, Germany; loredana.voelker-pop@anton-paar.com

**Keywords:** steady shear, transient shear, shear relaxation time, anisotropic colloids

## Abstract

Today, graphene oxide (GO) has gained well-deserved recognition, with its applications continuing to increase. Much of the processing of GO-based devices occurs in a dispersed form, which explains the commercialization of GO suspensions. Aging of these suspensions can, however, affect the shelf life and thus their application potential. Aging of GO preparations is often acknowledged, but no longer-term systematic study has been reported on the alteration of GO suspensions. This paper investigates high-concentration (10 mg/mL) aqueous GO suspensions over a 2-year time scale. In addition to steady shear tests, the dynamic behavior of the suspensions was studied in more detail by transient shear and frequency sweep measurements. Both the viscosity and the dynamic moduli increased with age, particularly within the first year. The results of the complementary Raman spectroscopic studies indicate that the change in the rheological behavior with aging results from a slow oxidation process occurring in the highly acidic aqueous medium during the relatively long-term storage. The (over)oxidized layers peel off spontaneously or are removed by high shear stress, resulting in increased viscosity, as it was corroborated by XRD and XPS.

## 1. Introduction

For many years, graphene oxide (GO) has been mainly considered as an intermediate of wet chemical graphene production. Today, however, graphene oxide itself has gained its own well-deserved recognition and its application potential is continuously expanding. Although in the manufacture of GO-based electrodes and optoelectronic devices, GO is used in a solid, dry form, most of the processing happens in a suspended form [1,2]. Direct ink writing (DIW), 3D printing, film casting, wet spinning, etc., are techniques that also require GO dispersed in liquid medium [3,4,5]. For this reason, GO is also commercialized in the form of aqueous dispersions.

Although synthesized for the first time 160 years ago [6], GO still remains to this day an elusive material whose properties are challenging to control. The performance of the nonstoichiometric nanoplates is influenced by multiple, occasionally interrelated factors, not only in the dry state but even more so in the suspended form. The list starts with the precursor graphite material, its origin (natural or synthetic), particle size, crystallinity or inorganic impurities [7,8]. It continues with the disintegration method of the graphite. Since Brodie’s pioneering work [6] several wet exfoliation techniques have been developed, including that of Staudenmaier [9] or Eigler [10] (a selected list is given in [1]), but the Hummers method [11] and its variations are probably those most frequently employed [4,5,12,13,14,15,16,17,18]. The oxidative wet exfoliation route including the nature of the purification steps determines both the morphology and the surface chemistry of the GO platelets [7,19]. The purified GO suspensions contain thin GO particles with a wide distribution of dimensions along the layer planes and the crystallographic *c*-axis: the GO suspension is a mixture of particles with diverse numbers of layers and aspect ratios. Surface chemistry involves the ratio of partly saturated and aromatic regions, carbon vacancies as well as the kind and distribution of the various oxygen functionalities along the basal plain and at the edges [20] affecting the interaction between the particles and with dispersing media.

The microstructure of the GO–water colloid system develops under the competition of long and short range, permanent and induced interactions evolving between the hydrated particles, hindered or fostered by the presence of water molecules. The complexity of interactions, including van der Waals interactions (e.g., π–π-stacking, dipole–dipole interactions), hydrogen bonding, or electrostatic interactions, results in a 3D network with a complex phase diagram, encompassing isotropic fluid, glass, gel and liquid-crystalline states [21,22]. At low concentration, isolated GO platelets are statistically distributed in the water and their motion is not correlated. At a critical concentration ϕ_c_ the percolation threshold is achieved and the sheets start to touch each other. The value of ϕ_c_, as well as the structures observed above, vary greatly in the literature [22,23,24,25,26,27]. The discrepancy is not surprising, as the aqueous GO systems studied by the various laboratories are intrinsically very different in several properties such as particle size distribution, aspect ratio or surface chemistry. In addition to particle properties, ions present in the aqueous medium can lead to different configurations of the GO sheets depending on their concentration and ionic strength [28,29]. H_3_O^+^ and consequently pH play a notable role among the ions affecting the internal structure [22,30]. The arrested state of concentrated GO suspensions can vary from repulsive glass (low ionic strength) to attractive glass or gel (high ionic strength) resulting in completely different rheological properties [21,31,32].

As noted above, GO is often processed in aqueous dispersion, and therefore in practical applications its rheology is of vital importance (Appendix A). Although the role of GO concentration is widely studied [2,3,23,24,26,27,33,34,35,36], the effect of the degree of oxidation of GO [37], the aspect ratio [22,38] and the temperature [27] on the rheological behavior has been also recognized by several groups. Since the graphite precursor and the exfoliation method affect the properties of GO [7,8,12,19,27], they could also influence the rheology of GO dispersion. A summary of the papers published on the rheology of concentrated aqueous GO suspensions is given in Appendix A. It is of utmost importance that all these papers study a single suspension prepared from a specific GO [3,22,23,26,27,38,39,40,41,42,43]. Although several publications comment on the alteration of graphene oxide with time, systematic works reporting these changes either in dry or suspended forms of graphene oxide are few. It was found that dry multilayer GO films undergo chemical reduction when stored at room temperature for a couple of months [44,45]. The publications are more contradictory about suspensions. In their early work, Titelman et al. already acknowledged changes in the pH of the dispersion medium but not in the interlayer distance and crystallite size in a 1.66 wt% aqueous GO suspension after 1 month storage [46]. Other groups also recognized minor changes owing to redox reactions between GO and water on a 1–2-month time scale: oxidation [47] or reduction [48], while high stability was reported by Chlanda et al. [49].

Aging of GO preparations is often acknowledged, but the few available results—as pointed out above—are contradictory. In order to fill this gap and get information about the shelf-life of aqueous GO suspensions, the rheological behavior of 10 mg/mL suspensions was investigated. This concentration exceeds the percolation threshold and is considered “high”. [21,23,31,38,50]. The rotational mode was applied to investigate the flow and viscosity curves. The dynamic behavior was studied both in oscillatory mode and under transient shear conditions. The change of the microstructure with aging and the effect of the mechanical stress were also explored by Raman spectroscopy.

## 2. Materials and Methods

### 2.1. Sample Preparation

Graphene oxide (GO) was obtained from natural graphite (Graphite Týn, Týn nad Vltavou, Czech Republic, average particle size 63 µm) by wet exfoliation following an “improved” Hummers’ method [11,12]. The product was washed thoroughly until a close to neutral pH of the supernatant (pH ~ 6) was achieved. Three batches were produced with a time gap of 1 year. The brown 0.72 *v*/*v*% (10 mg/mL) aqueous suspensions were stored in sealed amber bottles at ambient conditions (25 °C), avoiding agitation. Samples are distinguished as 0 y, 1 y and 2 y. All the rheology and Raman measurements were performed after obtaining the youngest (0 y) sample. He pycnometry of a freeze-dried probe yielded for the real density of GO provided a value of 1.39 g/cm^3^ (compared to 1.32 g/cm^3^ reported by Xu et al. [28]).

### 2.2. Characterization Methods

The morphology of the samples was characterized by a FEI Titan Themis 200 kV Cs corrected TEM with 0.09 nm HRTEM and 0.16 nm STEM resolution. The samples were drop-dried on TEM microgrids coated with an ultrathin carbon layer. Scanning electron microscopic (SEM) images of the gold-coated freeze-dried samples were taken by a JEOL JSM 6380LA (Jeol Ltd., Tokyo, Japan). X-ray photoelectron spectra were recorded on a Kratos XSAM 800 spectrometer operating in fixed analyzer transmission mode, using Mg Kα_1,2_ (1253.6 eV) excitation.

Raman spectra were recorded with a LabRAM (Horiba Jobin Yvon) instrument with a λ = 532 nm Nd-YAG laser source (15 mW laser power at focal point). Spectra were collected from at least three different spots in the circular quartz sample holder containing ca 0.1 mL suspension. After background correction, the first- and second-order regions of the spectra were deconvoluted into Lorentzian functions using the conventional fitting procedures of the Origin program. The Raman active lateral extension (*L*_a_) was estimated using the Tuinstra–Koenig–Cançado equation [51].
(1)$$L_{a}=\left(2.4\times 10^{-10}nm^{-3}\right)\lambda^{4}{\left(\frac{I_{D}}{I_{G}}\right)}^{-1}$$

λ is the wavelength of the laser source and *I*_D_/*I*_G_ is the intensity ratio of the D and G bands. Rheological studies were performed on a Physica MCR301 rheometer (Anton Paar, Graz Austria). All data were obtained using 25 mm-diameter parallel plate (PP25) geometry with a measuring gap of 300 μm. The temperature was maintained at 25 ± 0.01 °C using a Peltier device for the lower plate. For each measurement, about 0.5 mL of suspension was placed on the lower plate and the measuring gap was set. All measurements were performed in triplicate, using fresh samples in each case. Rheoplus V3.40 software was used for data analysis. Reliability of the data was checked for rotational and oscillatory tests by taking into consideration the torque limit of the measuring system and inertial effects of the measuring system and the samples as proposed by Ewoldt et al. [52]. Flow and viscosity curves were recorded in rotational mode. A preshear of 100 s^−1^ was applied for 30 s, then the shear rate was varied from 0.001 to 100 s^−1^ on a logarithmic scale with 5 data points in each decade. Each point was recorded for 10 s. Dynamic strain sweeps from 0.01 to 1000% were recorded in oscillatory mode at 1 rad·s^−1^ constant angular frequency. The range of the linear viscoelastic region (LVR) and the dynamic flow stress at the cross-over of the dynamic moduli were determined. Dynamic frequency sweeps from 500 to 0.05 rad·s^−1^ were recorded at a constant strain of 0.1% within the LVR, measuring 5 data points per decade. The relaxation spectra were calculated by the built-in edge-preservation method of the software. Transient measurements were made by applying a constant high shear rate (100, 1000 or 10,000 s^−1^) to the sample for 60 s in rotational mode followed by a constant shear at 0.1 s^−1^ to study the relaxation process. To follow the microstructural changes, polarized-light-imaging technique was applied to a MCR302 rheometer (Anton Paar, Graz Austria) equipped with a Peltier temperature-controlled glass plate and hood at 25 °C. The sample was illuminated with polarized light deflected towards the sample with the help of a beam splitter. The image of the illuminated sample was transferred telecentrically through an analyser filter to a CCD chip (1392 × 1040 pixels, 15 fps).

## 3. Results and Discussion

### 3.1. Characterization of the Fresh Sample

The main characteristics of the samples are illustrated in the case of the freshly prepared freeze-dried “2 y” sample. TEM images reveal that the particles are heterogeneous both in thickness (layer number) and in lateral extension (Figure 1).

The particle size changes over a wide range, from about 2 × 2 nm to ca 9 × 2 μm, which implies a variation of the lateral aspect ratio in the range 1–4.5. For the freeze-dried sample, the XRD analysis indicates an average height (*L*_c_) of about 11 nm and interlayer spacing of 0.77 nm. A similar interlayer spacing, 0.78 nm, was reported by Iakunkov et al. [53]. However, the dimensions are influenced by the nature and size (distribution) of the precursor graphite flakes as well as by the wet exfoliation method employed.

The Raman spectrum of the freeze-dried GO suspension was recorded in the range 800–3500 cm^−1^ [54]. In the first-order region of the Raman spectrum beside the D* (∼1220 cm^−1^, disorder from graphitic lattices and impurities), D″ (∼1506 cm^−1^, amorphous phases) and D′ (∼1600 cm^−1^, disordered graphitic lattices) bands, the well-known D (∼1350 cm^−1^, defect band, associated with structural disorder) and G (∼1580 cm^−1^, graphitic band, related to the vibrations of the sp^2^ building blocks) bands can be distinguished [55,56,57]. The second-order region is assigned to the 2D (∼2700 cm ^−1^, structural order) and the combined G*, D + D′ and 2D′ bands in the 2300–3800 cm ^−1^ region. In dry single-layer graphene samples, the G and 2D bands usually appear at 1585 and 2679 cm^−1^. Their shift to lower and higher positions, respectively, is a sign of increasing layer numbers. The 2D/G intensity ratio is also sensitive to the number of layers (>1.6 in single layer, dropping to 0.07 for four layers) [58].

### 3.2. Rheology

The processing and storage of concentrated GO suspensions require knowledge of the viscoelastic behavior of the suspensions and its potential change over time and after mechanical stress. Although several papers already exist on the rheology of GO suspensions (mainly of low concentration) [18,21,24,29,31,34], the aging of high-concentration suspensions has not been studied, despite their various applications as coatings, 3D printing materials or the production of polymer composites. At sufficiently high concentrations, an isotropic–nematic phase transition occurs in water, in which the transition point depends on particle size, aspect ratio and external conditions such as pH and ionic strength [1,2,17,21,22,28,29,30,31,33,34]. We note that the lowest concentration at which the biphasic or nematic region is reached varies largely in the literature, but a general range of 0.1 to 1 *v*/*v*% can be identified [21,23,31,38]. If attractive interactions are weak, the appearance of the nematic phase is entropy-driven [1,2], but lowering the pH or increasing the ionic strength of the solution results in weakened repulsive interactions and the contribution of the attractive interactions gradually increases, which in turn reduces the threshold concentration for phase transitions [59]. In dynamic rheological measurements a nonzero storage modulus develops as the particles, owing to their interactions, form a stress-bearing network. The interactions determine the arrested state of the GO suspension. Earlier, a glassy state was proposed at low salt concentrations that can melt upon addition of solvent. This state is explained by the repulsive electrostatic forces between GO platelets. At high salt concentrations, an attractive gel can form due to the dominant van der Waals interactions and screening of the electrostatic repulsion. In both states, birefringent structures can be observed [21].

Figure 2 shows the viscosity η and flow curves of the freshly prepared GO suspension (a representative viscosity curve with longer acquisition time and the low-torque limit are shown in Appendix A). Our measurements made on the samples with preshear conditions were highly reproducible, which indicates an even distribution of the particles in the concentrated suspensions (Appendix A). Pronounced shear-thinning behavior was observed over the whole shear rate range (Figure 2A) as the increase of shear rate causes shear alignment [21,27,31], while the flow curves clearly show the presence of a yield stress at low shear rates (Figure 2B). The flow-induced alignment of GO was demonstrated by Yang et al. using SEM images under flow [41]. The monotonic decrease in viscosity over the whole shear rate range and the absence of a plateau at high shear rates suggests the presence of an extended network that gradually breaks into smaller domains with increasing shear stress. This is consistent with the fact that the viscosity of the suspension is much higher than that of the solvent even at the highest shear rate used.

Kumar et al. [38], investigating the dependence of viscosity on GO volume fraction, found a nonmonotonic relationship, which they explained by the formation of an LC phase or a birefringent gel phase. Similarly, although the values differ, possibly due to the different surface chemistry and/or aspect ratio of the GO particles, the formation of a birefringent structure was assumed in our case in view of the high viscosity values and the same trend in viscosity–shear rate curves. Therefore, we applied polarized light imaging to follow microstructural changes during shearing (Figure 2C). An assumed birefringent texture is observed at low shear rates (1 s^−1^), with rather isolated domains. At higher shear rates (10 and 100 s^−1^), birefringence spreads over larger areas due to the alignment of domains. Similar shear-thinning behavior was observed by Corker et al. with low-shear viscosity values [3], whereas Konkena et al. [21] obtained zero-shear viscosity values around 1000 Pa∙s. Both results were obtained in the concentration range also used in our work. Again, the dissimilar surface chemistry and particle size and distribution strongly affect the microstructure and thereby the viscosity curve. Finally, Bai et al. [22] studied the viscosity at different pH values and obtained low-shear viscosity values of the same order of magnitude (10^4^ Pa∙s) as in our work.

As yield stress was observed in Figure 2B, the measured data were fitted to three commonly used rheological models, namely the linear Bingham, the power-law and the nonlinear Herschel–Bulkley (HB) models [60] in the shear rate range of 0.1 to 100 s^−1^. Thus, the viscosity values at very low shear rates were not included, where the steady state is not completely reached, as demonstrated by the change of curve shape with increased acquisition time, visible in Appendix A. The shear stress *τ* can be quantified by the Bingham model, which assumes Newtonian behavior for shear stress values above the yield stress:(2)$$\tau =\tau_{0}+k\dot{\gamma }$$
where τ is the shear stress, $\dot{\gamma }$ is the shear rate, $\tau_{0}$ is the steady shear yield stress; and *k* is the consistency coefficient of the model representing the limiting viscosity of fluid at infinite shear rate. The power-law model describes the deviation from Newtonian behavior as a power law of the shear rate:(3)$$\tau =k\cdot {\dot{\gamma }}^{n}$$
where *k* is the consistency coefficient of the model and *n* is the flow index (0 < *n* < 1). The *n* values indicate the deviation from Newtonian behavior. For isotropic liquids with no long-term ordering, the value of *n* is close to 1, while shear-thinning liquids can be characterized with an *n* value well below 1. On combining the yield stress and the non-linear dependence of shear stress on shear rate, we obtain the HB model, which assumes non-Newtonian behavior above the yield stress
(4)$$\tau =\tau_{0}+k\cdot {\dot{\gamma }}^{n}$$
where *k* is the consistency coefficient of the model, and *n* is the flow index.

The Bingham model describes the flow curve poorly as it cannot accommodate for the nonlinearity of the function. Although the power–law model performs better, deviation is encountered both at low and high shear rates. The corresponding fits are shown in Appendix A. The HB model, which takes into account both the yield stress and the nonlinear shape, yielded the highest quality fit (Figure 3, Table 1). The good quality of the fits indicates that both yield stress and nonlinear behavior are important and must be considered on comparing samples.

As noted in Appendix A, the steady shear viscosity of “concentrated” GO suspensions has been studied across the literature. Nevertheless, for a deeper understanding of the rheological behavior, oscillatory experiments are required. In this method a sinusoidal strain is applied to the sample and the time-dependent stress is recorded, providing two basic pieces of information: the amplitude of the response (i.e., the stress developed under the applied strain) and the phase lag between the strain and the stress. These are usually converted into two dynamic moduli. The storage modulus *G*′ represents the elastic response of the material and is equal to the energy density stored in the material during deformation, while the loss modulus *G*″ describes the viscous character and is related to the energy dissipated during deformation. The ratio of the two moduli *G*″/*G*′, i.e., the ratio of the viscous to the elastic property (also known as damping factor), contains important information on the existence of long-range structure.

The effect of strain at constant angular frequency on the dynamic moduli is shown in Figure 4. To determine the linear viscoelastic region, a wide strain range was used, from 0.01 to 10,000%. Consistent with the literature [23,38], almost constant moduli were found up to around 1% strain. Above this critical value, both moduli became stress-dependent with a moderate drop of the loss modulus in the strain range 1–10,000% and a reduction in storage modulus from approximately 2000 Pa to 1 Pa. This huge decrease is explained by the disintegration of the network under shear into oriented GO flakes. The elastic nature of the network becomes small at high strains. The cross-over point of the moduli can be used to characterize the flow stress in dynamic mode. The dynamic flow stress is about one order of magnitude greater than that reported in previous papers (e.g., about 10 Pa in [23] and [38]), but, as mentioned above, the surface chemistry and aspect ratio (distribution) of the GO particles could explain the difference. The flow stress determined from the oscillatory shear is considerably higher than the yield stress from steady shear measurements, but it should be recalled that the two stress values were determined under different types of shear load.

The viscoelastic nature of the samples was characterized by following the dynamic moduli over a wide range of frequencies within the range of linear viscoelasticity (0.1% strain). As shown in Figure 5, the storage modulus (a few 100 Pa) exceeds the loss modulus (by a few 10s of Pa) below 100 rad·s^−1^, above which the dynamic moduli approach each other. The storage modulus is independent of frequency, while the loss modulus is slightly influenced by the frequency, with a pronounced increase at frequencies higher than 10 rad·s^−1^. This behavior is typical of structured materials (i.e., viscoelastic solids) with nonzero storage modulus at low frequencies [21,23,27]. This means that the elasticity of the network becomes independent of time at very long timescales. In other words, the relaxation of the network cannot be accessed in the experimentally available time window. Furthermore, the increase in loss modulus at high frequency implies the presence of a fast relaxation process, which, owing to their limited frequency range, was not detected by Konkeva et al. [21]. Our modulus values are, however, in agreement with their results, suggesting the presence of the gel phase that was reported in their work around a concentration of 0.9 *v*/*v*%. The same viscoelastic behavior was observed by Naficy et al. [23] above 4.5 mg/mL, but the change of frequency dependence at high frequencies was not interpreted.

To reveal the dominant relaxation processes, the frequency spectra were further analysed by the Cole–Cole representation and by plotting the damping factor (Figure 6). The Cole–Cole representation of the dynamic moduli gives a qualitative picture of the relaxation processes under dynamic load. The presence of a single semicircle would indicate one main relaxation process. Here, a complete semicircle was not observed (Figure 6A) due to the limited frequency range available. (To avoid evaporation during the long measurement, very low frequencies cannot be measured, and meanwhile the instrument has an upper frequency limit.) Nevertheless, the two (albeit partial) semicircles imply at least two separate relaxation processes. The frequency-dependent damping factor (Figure 6B) also confirms the existence of two main processes, since a minimum is observed in the medium-frequency range with an increase on either side at low and high frequencies.

A general relaxation time spectrum (relaxation function, *H*(λ) as a function of relaxation time, λ) was calculated from the frequency sweep measurements (Figure 7). Although the resulting spectrum should be treated with caution due to its sensitivity to experimental error, a slow relaxation process in the 10 s range and at least one fast process (10 to 100 ms) can be distinguished. The very fast relaxation implies that after shearing of these concentrated GO suspensions, their structure can reorganize quickly. The existence of different relaxation times is known for colloidal glasses such as Laponite^®^ [61]. In that case, the short relaxation process is attributed to small displacements of the platelets within the cage of their neighbouring particles, while the slow mode corresponds to escape of the platelets from their cage.

To study the effect of the high shear load on the rheological properties and the microstructure, transient measurements were performed in rotational mode. Shear rates of 100, 1000 and 10,000 s ^−1^ were subsequently applied with a 60 slow-speed (0.1 s^−1^) rotational shear (Figure 8A). The steady-state was quickly achieved at shear rate 100 s^−1^ and a very fast recovery of the viscosity was observed after returning to 0.1 s^−1^. The viscosity measured at this low shear rate is in good agreement with that observed in the original suspension, indicating complete recovery of the microstructure and the reversible nature of the deformation. This also implies the absence of rejuvenation, as observed in Laponite^®^ systems. Interestingly, the shear at higher rate (1000 s^−1^), which exceeds the inverse of the fast relaxation time (10–100 ms), produced a very different effect. First, the viscosity decreased under shear, but then an increase by approx. 50% of the initial value of this regime was observed at constant shear rate. During the relaxation period (at 0.1 s^−1^) the suspension displayed increased viscosity with respect to that after the previous low-shear step.

At the highest rate investigated, 10,000 s^−1^, the viscosity did not change after the initial drop, but during the relaxation the final viscosity increased even further than after the 1000 s^−1^ step. The viscosity curves after these transient measurements were recorded again, and although the shape of the curves did not change, enhanced viscosity was observed in the whole shear rate range (Appendix A). This indicates that the nature of the microstructure did not in general change with the high-rate shear, but some irreversible structural change affecting the flake size (distribution) and/or aspect ratio nevertheless occurred. As was shown already for concentrated GO dispersions, it is known that anisotropic colloidal particles tend to form liquid crystalline structures [31]. The ratio of the nematic (ordered) and isotropic phases is influenced by the concentration, size and aspect ratio of the particles, as well as by the ionic strength of the medium [21]. In our system, the birefingance gel present in the initial state might convert into a liquid crystal upon high shear followed by the breakage of ordered domains and alignment of smaller particles in shear direction (Figure 8B). At rest, the microstructure is not completely ordered, and we assume that GO remains in randomly oriented clusters, as proposed by Giudice et al. [38] The strong shear-thinning at low shear (100 s^−1^) can be explained by partial orientation and ordering of the particles, as shown in the upper panel of Figure 9 [21,23,27,31].

When the external constraint suddenly stops, this temporary state returns to the original, more random orientation of particles. At higher frequency (Figure 9, lower panel), however, we propose a different scenario. It was reported that in aqueous suspensions, the GO layer distance increases to as much as 1.2–1.3 nm, as water penetrates into the interlayer space [62], thereby weakening the π–π integrity of the layered structure. Moreover, the GO flakes contain highly functionalized external regions. Even if they are only partly deprotonated, their repulsive nature further weakens the attractive interaction between the graphenic layers. These effects lead to removal of the external layers from the platelets under high shear forces. As a consequence, new surfaces are born that adsorb water from the bulk phase, finally resulting in loss of free bulk water and an increase in viscosity. For this reason, the process at high shear rate is considered to be irreversible as the enhanced viscosity and the smaller hydrated debris hinder the mobility of the particles. The structural change is corroborated by the alteration of birefringence at rest after 1000 s^−1^ and 10,000 s^−1^ with gradual size reduction of the birefringent units and development of a more homogeneous structure (Figure 8B). Freeze-dried samples were prepared from the suspensions before and after high shear for SEM imaging. The images in Figure 10 confirm the microstructural change caused by the mechanical treatment. The loose structure of the GO becomes denser and somehow more regular after the mechanical treatment.

Table 2 compares the parameters derived from the X-ray diffractograms of the same samples. The change of the peak position and shape already marks the morphological changes. The average number of the layers halved after the mechanical treatment, an obvious mark of peeling off.

### 3.3. Aging

The microstructure of a concentrated suspension can evolve over time [63]. GO is also commercialised in the form of suspensions to meet the demand for various applications. It is therefore of crucial interest to know if the shelf-life affects the properties of the aqueous GO suspensions. The brown 0.72 *v*/*v*% (10 mg/mL) aqueous suspensions were stored in sealed amber bottles at ambient temperature (25 °C), avoiding agitation. The change in rheological characteristics was carefully checked by studying three batches of GO suspensions of similar concentration, produced with a time gap of 1 year and distinguished as 0 y, 1 y and 2 y, respectively. All measurements were performed in triplicate on all three batches at the same time. For brevity, one of the parallel measurements is shown in the main text and the complete data sets are listed in the Appendix A.

We mentioned earlier that the parameters of the HB model completely characterize the viscosity curve. Instead of the zero-shear viscosity used in the literature [3,23], we therefore used them to compare the suspensions of different ages (Table 3). The shape of the flow and viscosity curves did not change over time (Appendix A). An increase in yield stress and viscosity values was observed after 1 year of storage, but not after an additional year. This indicates that the development of the microstructure takes place mainly within the first year, and remains stable during further storage.

The evolution of the microstructure was further analysed by oscillatory measurements. Strain sweep measurements show an increase with storage time of both dynamic moduli (Appendix A) but the strain at the dynamic moduli cross-over remained the same, indicating that the microstructure did not change generally. Similar to the moduli, the dynamic flow stress also increased during aging and more than doubled after storage (Table 3). The similarity in dynamic behavior of the GO suspensions of different ages is also captured by the frequency sweep measurements (Appendix A). Here again, the tendencies are the same for all samples, but an increase of the moduli is observed upon aging. The relaxation processes are visualized in the Cole–Cole plots of the three samples of different ages (Appendix A), where the same semicircles for the two main processes can be identified. The damping factor shows the same trend as before regardless of age, although a very slight reduction indicates a corresponding relative increase of the elastic nature of the samples (Appendix A).

The characteristic relaxation times deduced from the relaxation spectra indicate that the fast relaxation time does not change with aging (Figure 11). With aging in Laponite^®^ glasses, the slow mode can shift to longer times owing to the increase of the cage size according to the evolution of the microstructure [61,64]. Similarly to Laponite^®^ suspensions, in our case the slower process appears at slightly shorter relaxation times, thereby allowing the observation of an even slower relaxation that shifts into the detection window (between 10 and 100 s). Both the modest acceleration of the slower processes and the increased dynamic flow stress could be the sign of alteration of the GO flakes.

The transient experiments reveal the same fast relaxation for samples of different ages (Figure 12) and the strengthening of the samples upon high-shear treatment (1000 and 10,000 s^−1^). This behavior differs from that of soft glasses, e.g., Laponite, in which shear rejuvenation was reported, involving reduction in viscosity, followed by slow relaxation [61,64].

Considering the similarity in shape of the viscosity curves before and after the high shear treatment of the samples of different ages, it is reasonable to assume that the mechanism of the microstructural change during storage is similar to the process shown in Figure 9. The effect of the idle time and the highest shearing on the HB consistency index *k* is shown in Table 4.

By the end of the first year, the consistency index increased to a value similar to that reached at high-rate shearing on fresh samples. Later, the *k* value of the samples did not change with time, but the mechanical treatment can still produce an increase in the consistency index. The results indicate that after the first year, no further aging is observed in rotational tests during the extended storage, but external forces can induce microstructural changes, which are enhanced with idle time.

To reveal the transformation in the suspensions during the storage the batches of different ages were also studied by Raman spectroscopy (Figure 13). In the first-order region the deconvolution was performed similarly to the dry sample, but the second-order region was considered only up to 3000 cm^−1^ (“losing” the D + D′ and 2D′ bands) due to the strong signal from water in the higher Raman shift region. The best fit was found when the D* band, which is reportedly related to disordered graphitic lattices ensured by sp^2^–sp^3^ bonds in the first-order region, was neglected [65]. The peaks were characterised by their position and their intensity related to the G band (Table 5). In their pioneering work, Tuinstra and Koenig established a relationship between the *I*_D_/*I*_G_ ratio and the lateral size of the graphenic platelets. This formula, however, should be used with caution, and only as a first approximation. Based on the experimental data of Beny-Bassez and Rouzaud [66], Cuesta et al. pointed out that this estimation depends strongly on the nature of the sample [67] and is defined rather by the Raman active defects than by the “physical” extension of the platelets [51]. As the difference in rheological properties between the samples is amplified by shearing, samples of different age were also investigated before and immediately after the shearing experiment. The full set of data derived from the deconvoluted spectra is shown in Table 5. The effect of aging was tracked by studying the D, G and 2D bands.

While the position of the D and G bands remain constant, only the 2D exhibits a systematic increase, i.e., a blue shift on going from year 0 to year 2. The wide 2D peaks indicate that the particles contain several layers and the planes may be randomly oriented. The intensity ratio *I*_D_/*I*_G_ is the hallmark of the defects, i.e., the extent of the sp^2^ domain size of the graphene sheet containing sp^2^ and sp^3^ carbons. It was reported earlier that chemical reduction decreases the *I*_D_/*I*_G_ ratio through restauration of the sp^2^ states of the carbon [58]. In the batches of different ages, the limited but systematic concomitant increase of the *I*_D_/*I*_G_ and the *I*_2D_/*I*_G_ ratios indicates a slow oxidation process. During long-term storage, the dissolved oxygen present in the intrinsically acidic medium can act as an oxidizing agent. Oxidation can further reduce the π-π interaction between adjacent layers by enhancing the repulsion along the edges and lead to gradual deterioration of the already oxidized graphene sheets at the peripheries. These processes also result in a systematic decrease of the average lateral dimension.

Mechanical stress has only a minor effect on the position of the G and 2D bands. The shift of the G and the 2D bands into higher and lower wavenumbers could be a sign of decreasing layer numbers [58]. Except for the fresh sample, the 1 y and 2 y batches show a drop in the *I*_D_/*I*_G_ and *I*_2D_/*I*_G_ ratios, implying an enhancement of the structural order. The increase in lateral dimensions seems to confirm that the slow oxidation corrupts the integrity of the external sheets. Mechanical force can remove these (over)oxidized layers, leaving behind particles with a higher order on average. The reappearance of the D″ peak could be related to the amorphous part that is removed.

The oxidation during aging was also confirmed by X-ray photoelectron spectroscopy (XPS). Probes from the “2 y” sample were taken and freeze-dried at various times for analysis. In the fresh young sample, the O/C atomic ratio was 25/75. After 1 year it was 30/70 [54], and when measured after 2 years we found 33/67, corroborating the oxidation detected by Raman spectroscopy. The C1s and O1s regions were decomposed according to [68]. The oxidation process can be clearly followed by the alteration of the various C1s and O1s species in Figure 14. Although the changes are spectacular, the trend of the changes is additionally shown in Appendix A. The changes also were monitored by the pH, which dropped from 6 (as prepared) to 2.2 after 3 years, following a similar tendency reported by Titelman [46].

## 4. Conclusions

For these rheological studies, three concentrated (10 mg/mL) aqueous graphene oxide suspensions were prepared, each with a 1-year gap, by the improved Hummers wet exfoliation method. The steady shear rotational tests revealed strong shear-thinning behavior, which is explained by partial orientation of the GO flakes in the shear force field. After relaxation at low shear, the initial viscosity values were recovered, thereby implying a reversible change in the structure at moderate shear rates (≤100 s^−1^). The rotational tests in oscillatory mode detected a well-defined yield stress. The existence of yield stress is due to the GO particles touching each other and thus restricting their own mobility. The dynamic behavior of the suspensions, studied by frequency sweep tests, displayed relaxation modes of the suspensions on different time scales. A very fast relaxation in the millisecond time range was detected, the existence of which is further substantiated by the plateau viscosity, which, in the transient shear measurements, was reached within a short time. Interestingly, an irreversible increase of viscosity was observed at high shear rates (≥1000 s^−1^) for which we observed a microstructural change and showed that is related to a process in which the highly oxidized external GO sheets are peeled off from the flakes. Both the viscosity and the dynamic moduli increase with age, particularly within the first year. The aged samples display similar shear-thinning behavior in rotational tests, and frequency-dependent dynamic moduli in oscillatory tests, but the characteristic relaxation times decrease slightly. This again suggests a microstructural change, similar to that under high-shear load. The results of the complementary Raman spectroscopic studies indicate that the alteration in the rheological behavior with aging is the result of a slow oxidation process that occurs in the highly acidic aqueous medium during long-term storage and leads to spontaneous peeling of the external sheets from the flakes. Our observations draw attention to the role of storage time and mechanical load in the performance of such concentrated GO suspensions that must be considered in processing technologies and applications.

## Figures and Tables

**Figure 1 nanomaterials-12-00916-f001:**
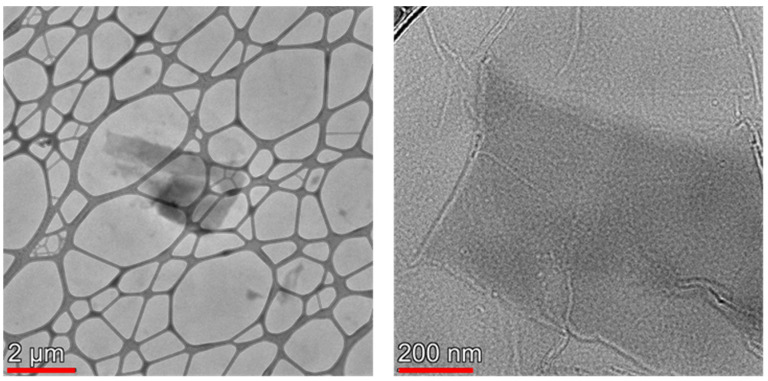
TEM images of the GO particles. The scale bar is 2 μm and 200 nm on the left- and right-hand side, respectively.

**Figure 2 nanomaterials-12-00916-f002:**
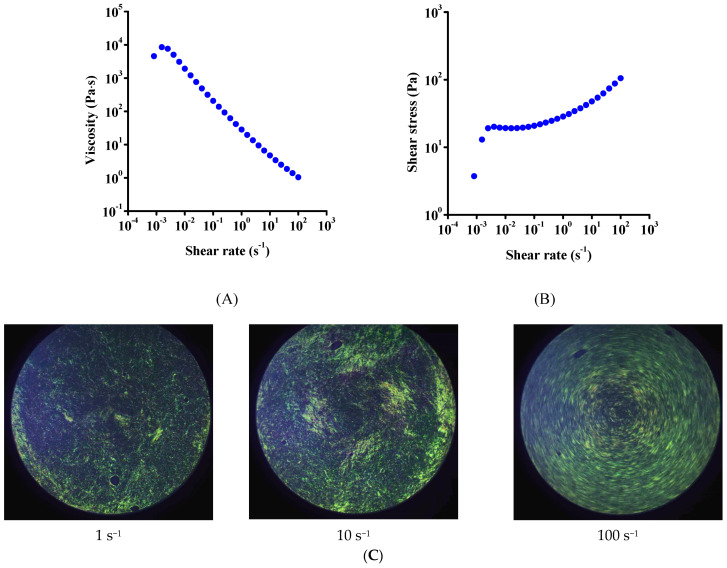
Steady shear measurements of fresh suspensions. Effect of shear rate on viscosity (**A**) and shear stress (**B**). Measurements were performed in triplicate using fresh samples in each case. The complete datasets are shown in Appendix A. In situ polarized light images of GO suspensions at different shear rates (**C**).

**Figure 3 nanomaterials-12-00916-f003:**
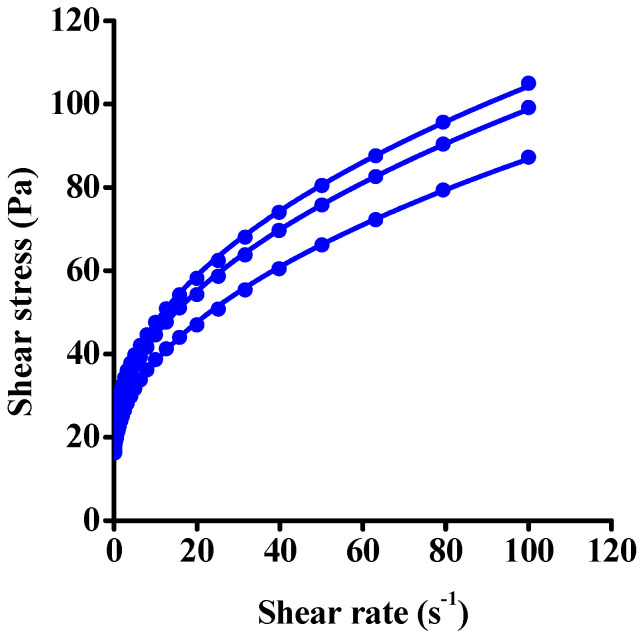
Experimental shear stress–shear rate data obtained on fresh sample (symbols) and their fit to the Herschel–Bulkley model (lines).

**Figure 4 nanomaterials-12-00916-f004:**
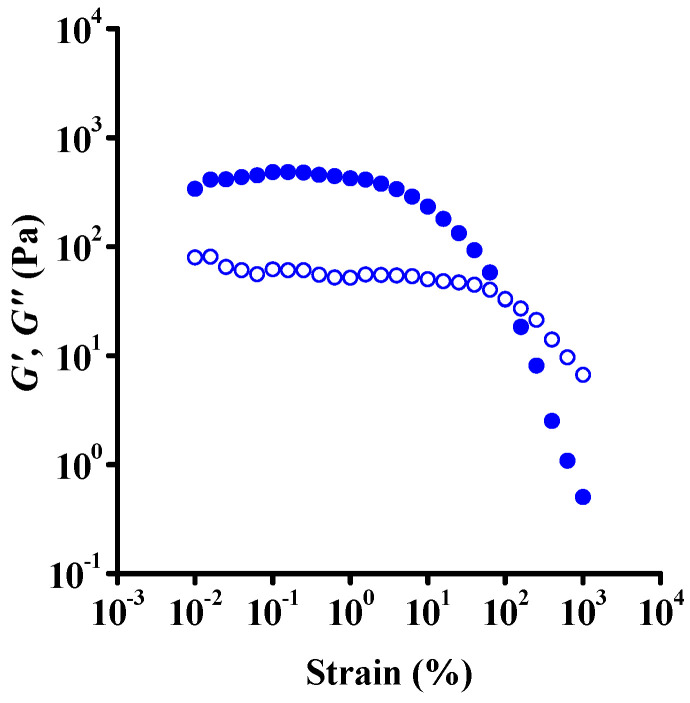
Strain-dependent oscillatory shear measurements on a fresh suspension at constant angular frequency 1 rad·s^−1^. Solid and open symbols represent storage (*G*′) and loss (*G*″) moduli, respectively. Results of the parallel measurements and the low-torque limit of moduli are shown in Appendix A.

**Figure 5 nanomaterials-12-00916-f005:**
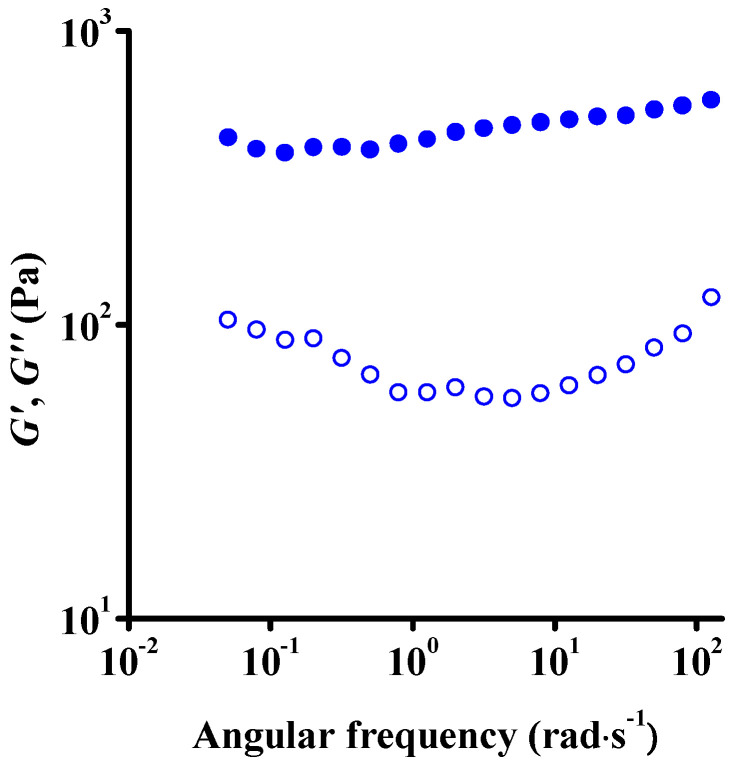
Dynamic frequency sweep of the fresh suspension at constant strain amplitude 0.1%. Data points above 100 rad·s^−1^ are of an approximate nature due to the instrument limits. Solid and open symbols represent storage (*G*′) and loss (*G*″) moduli, respectively. Results of the parallel measurements and inertial limits of the measurements are shown in Appendix A.

**Figure 6 nanomaterials-12-00916-f006:**
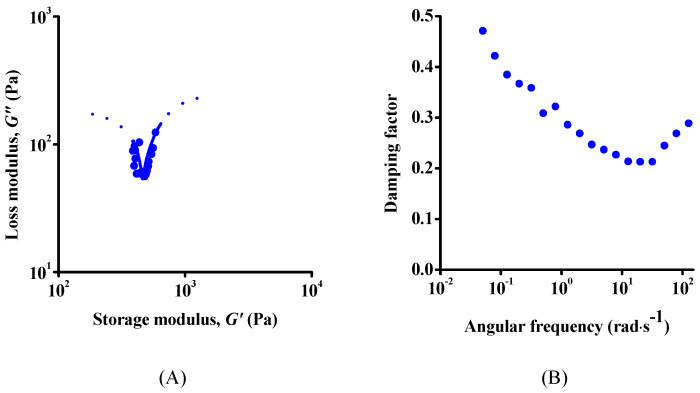
Cole–Cole representation (**A**) and the frequency dependence of damping factor (**B**) of fresh suspension.

**Figure 7 nanomaterials-12-00916-f007:**
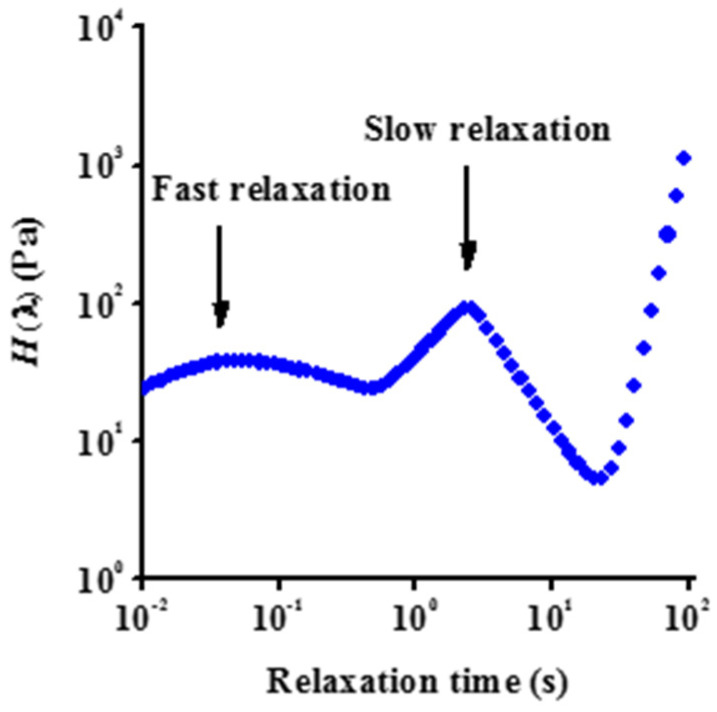
Relaxation time spectrum of a fresh aqueous suspension.

**Figure 8 nanomaterials-12-00916-f008:**
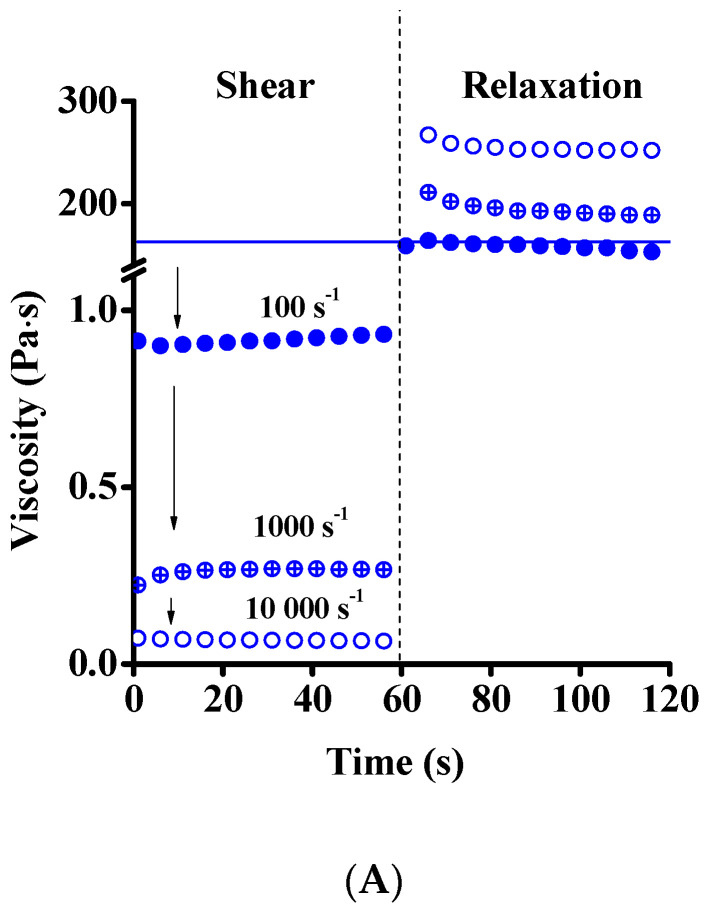
Change of viscosity of fresh GO suspensions under high shear followed by relaxation at low shear rate (**A**). Three consecutive experiments were performed with a shear rate of 100, 1000 and 10,000 s^−1^ for 60 s in each case in the first part, and 0.1 s^−1^ for relaxation (60 s). Polarized light images of GO suspensions at rest after shearing with different shear rates (**B**).

**Figure 9 nanomaterials-12-00916-f009:**
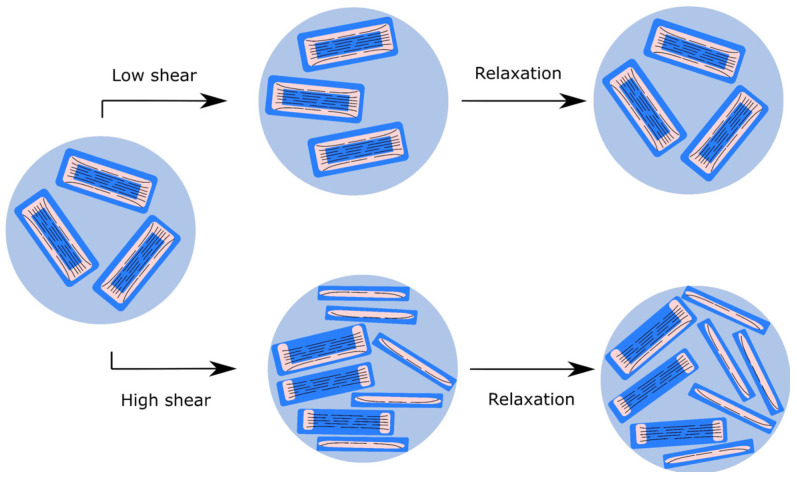
Proposed mechanism for microstructural changes of GO suspensions upon shearing and relaxation. Black lines represent GO platelets, pink areas strongly functionalized regions with high oxygen content, dark blue is bound water, light blue shows the bulk aqueous phase.

**Figure 10 nanomaterials-12-00916-f010:**
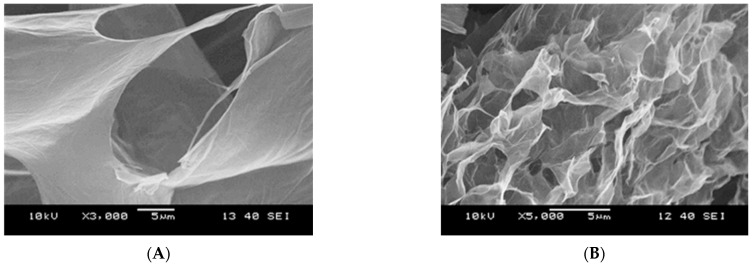
Scanning electron micrographs of GO samples freeze-dried from 10 mg/mL suspension before (**A**) and after shearing at 10,000 s^−1^ for 5 min (**B**). The sheared sample was transferred to a plastic tube immediately after the mechanical treatment and dropped into liquid nitrogen prior to freeze-drying. The scale bar marks 5 μm. The images were taken on the 1y sample.

**Figure 11 nanomaterials-12-00916-f011:**
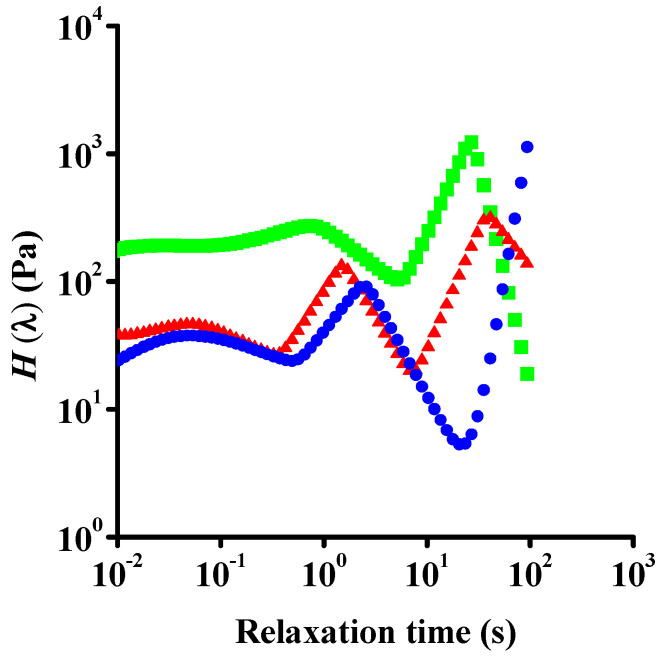
Relaxation process of aqueous GO suspensions of different ages. 0 y—blue dots; 1 y—red triangles; 2 y—green squares.

**Figure 12 nanomaterials-12-00916-f012:**
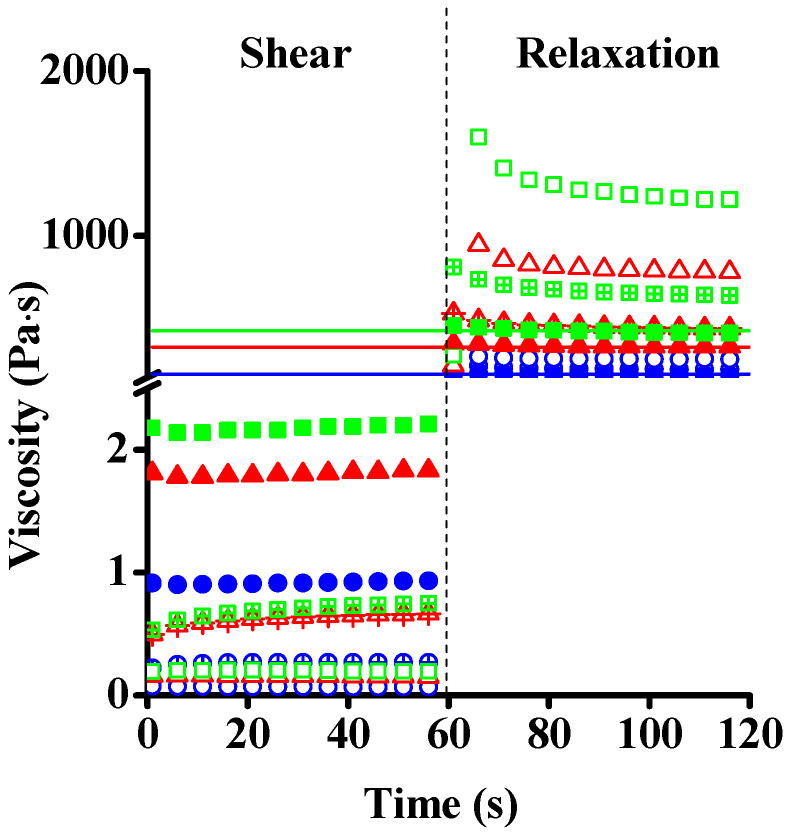
Change of viscosity of GO suspensions of different ages under high shear followed by relaxation at low shear rate. Constant shear rate of 100 s^−1^ (open symbols), 1000 s^−1^ (open symbols with a + sign) and 10,000 s^−1^ (filled symbols) applied for 60 s followed by a relaxation at 0.1 s^−1^. 0 y—blue dots; 1 y—red triangles; 2 y—green squares.

**Figure 13 nanomaterials-12-00916-f013:**
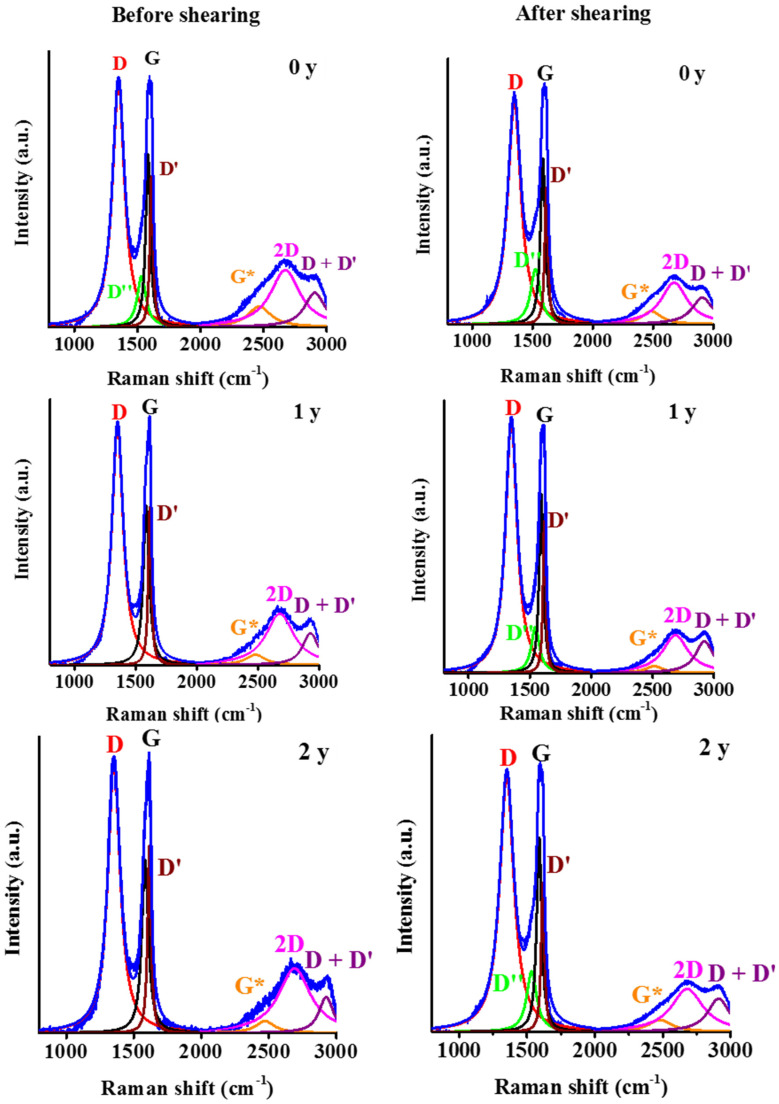
Deconvoluted Raman spectra of the samples of different ages before and after shearing.

**Figure 14 nanomaterials-12-00916-f014:**
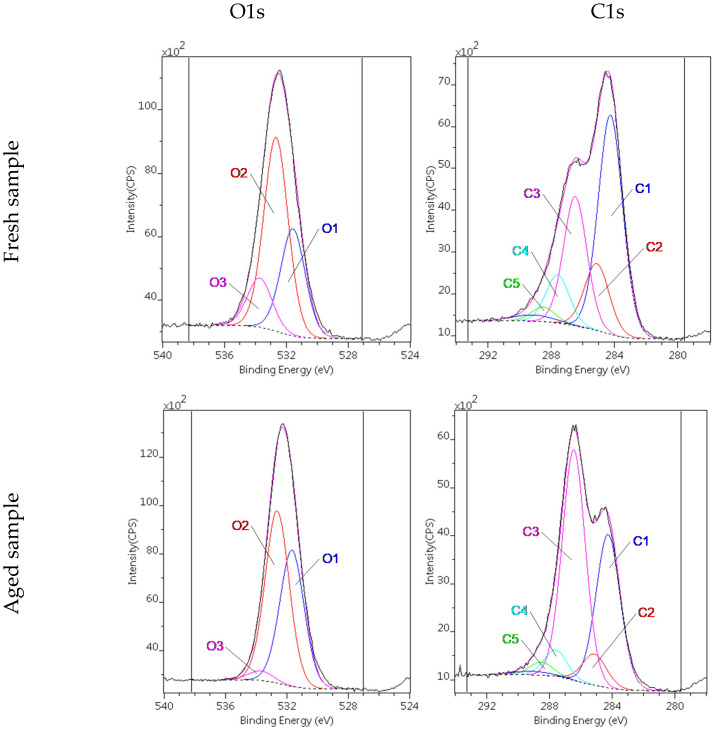
Decomposed O1s and C1s region of the fresh and aged GO samples. (Probes from the “2y” sample were freeze-dried and analysed both when it was fresh and after 2 years.).

**Table 1 nanomaterials-12-00916-t001:** Fitted parameters from the Herschel–Bulkley model.

Dataset	*τ* _0_	*k*	*n*	R^2^
	(Pa)	(Pa·s)		
1	14 ± 1	8 ± 0	0.49 ± 0.01	0.9996
2	19 ± 0	8 ± 0	0.50 ± 0.00	0.9999
3	20 ± 0	9 ± 0	0.49 ± 0.00	0.9999

**Table 2 nanomaterials-12-00916-t002:** XRD characteristics of freeze-dried graphene oxide before and after shearing *.

	Peak Position	FWHM	*d*	*L* _c_	*N* _c_
	(001)	(001)			
	(Degree)		(nm)		
Before shearing	11.5	0.65	0.77	12.8	17
After shearing	10.7	1.07	0.83	7.7	9

* FWHM—full width at half maximum of peak 001; *d*—interlayer spacing from Bragg low; *L*_c_—average height from Debye-Scherrer equation; *N*_c_—average number of layers.

**Table 3 nanomaterials-12-00916-t003:** Evolution with aging of Herschel–Bulkley parameters and dynamic flow stress.

Age of Sample	Parameters of HB Model *				Dynamic Flow Stress
	*τ* _0_	*k*	*n*	R^2^	τ
(y)	(Pa)	(Pa·s)			Pa
0	14 ± 1	8 ± 0	0.49 ± 0.01	0.9996	50.4
	19 ± 0	8 ± 0	0.50 ± 0.00	0.9999	42.1
	20 ± 0	9 ± 0	0.49 ± 0.00	0.9999	46.6
1	32 ± 1	22 ± 0	0.49 ± 0.00	0.9998	57.9
	33 ± 0	14 ± 0	0.50 ± 0.00	0.9997	68.5
	39 ± 1	15 ± 1	0.51 ± 0.01	0.9986	74.5
2	27 ± 1	16 ± 1	0.43 ± 0.01	0.9993	93.7
	35 ± 1	20 ± 0	0.47 ± 0.00	0.9997	115
	38 ± 1	18 ± 1	0.48 ± 0.01	0.9996	103

* *τ*_0_—steady shear yield stress; *k*—consistency coefficient; *n*—flow index.

**Table 4 nanomaterials-12-00916-t004:** Herschel–Bulkley consistency indices after high-rate (10,000 s^−1^) shearing.

Age of Sample	*k*
(y)	(Pa·s)
0	11 ± 0
	14 ± 1
	15 ± 1
1	35 ± 1
	24 ± 1
	27 ± 1
2	26 ± 1
	34 ± 1
	30 ± 0

**Table 5 nanomaterials-12-00916-t005:** Characteristic parameters deduced from the Raman signals.

Characteristics		Unit	Before			After		
			Shearing					
			0 y	1 y	2 y	0 y	1 y	2 y
Position	D	cm^−1^	1350 ± 1	1351 ± 0	1354 ± 0	1350 ± 1	1348 ± 0	1350 ± 1
	D″		1538 ± 1	-	-	1528 ± 2	1541 ± 2	1535 ± 0
	G		1585 ± 0	1585 ± 1	1584 ± 1	1587 ± 1	1592 ± 0	1589 ± 1
	D′		1611 ± 0	1615 ± 0	1615 ± 0	1616 ± 1	1617 ± 0	1616 ± 1
	G *		2465 ± 1	2486 ± 9	2473 ± 6	2475 ± 18	2519 ± 18	2486 ± 12
	2D		2671 ± 1	2680 ± 1	2688 ± 4	2673 ± 2	2686 ± 4	2681 ± 2
	D + D′		2905 ± 1	2929 ± 3	2921 ± 3	2911 ± 2	2921 ± 2	2911 ± 2
*I*_D_/*I*_G_		-	1.40 ± 0.00	1.53 ± 0.02	1.57 ± 0.02	1.35 ± 0.07	1.42 ± 0.02	1.33 ± 0.03
*I*_D″_/*I*_G_		-	0.30 ± 0.00	-	-	0.30 ± 0.00	0.25 ± 0.02	0.32 ± 0.02
*I*_2D_/*I*_G_		-	0.31 ± 0.03	0.30 ± 0.03	0.41 ± 0.04	0.23 ± 0.03	0.19 ± 0.02	0.22 ± 0.02
*L*_a_ *		nm	13.7 ± 0.00	12.6 ± 0.1	12.2 ± 0.2	14.3 ± 0.8	13.6 ± 0.2	14.5 ± 0.3

* from Equation (1).

## Data Availability

Numerical data are available on request from the corresponding author.

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
