# Peer review of "Long-Term Aging of Concentrated Aqueous Graphene Oxide Suspensions Seen by Rheology and Raman Spectroscopy"

_nanomaterials, 2022, doi:10.3390/nano12060916_

Round 1

Reviewer 1 Report

In this article, the authors focus on the effects of prolonged ageing on the dispersion properties of suspensions. The authors use a higher concentration of graphene oxide suspensions as an example, with an ageing time span of two years. Experiments with steady-state shear, transient shear and frequency sweep revealed a slow oxidation process in the graphene oxide dispersion during long-term storage and that the oxide layer may be exfoliated under high shear, leading to an increase in viscosity.
Overall, this manuscript focuses on one of the more important scientific issues, which is the perception of the evolution of nanoparticle dispersions with increasing placement time. The authors give some interesting conclusions based on their own research. At the same time, the authors also give a relatively rich and complete experimental evidence and a rigorous logical derivation.
In view of this, I recommend the manuscript for publication in your journal with minor revisions. Specific comments are as follows.
(1) Is there any other evidence for the peeling of the oxide layer under high speed shear as suggested by the authors? This is a very important finding.
(2) Is there a problem with the picture in Fig. 1?

Translated with www.DeepL.com/Translator (free version)

Reviewer 2 Report

This article describes the aging of GO suspensions. The rheology and Raman studies of GOO suspensions were carefully investigated. I suggest the manuscript for publication after considering the below points.

  • It will be good to compare the studies of all three samples.
  • TEM images of the 2Y sample are only shown, with this image it is difficult to conclude the surface morphology of GO. TEM images for 0Y, 1Y and 2Y are needed. Also, the scale bar in Figure 1 is missing.
  • Rheology for 1Y and 2Y is missing.
  • Raman studies before and after looks almost the same?
  • Please explain the changes in morphology with aging using TEM and SEM images.
  • Did the authors check the concentration of GO suspension with aging?
  • Did the authors separate the upper layer for the shear stress or use the whole suspension?
  • If separated by aging usually viscosity will decrease for the upper layer where thin layers of graphene will be present.

Author Response

Pleasesee attachment.

Reviewer 3 Report

The authors presented a rheological sudy of aged GO suspensions up to 2 years. The subject is of interest nd of increasing application, so the results can be helpful for the area. Therefore, the manuscript should be considered for publication, after addressing some issues detailed below.

  1. The results are obtained for only one GO suspension. I wonder how general are the results (regarding, e.g. the influence of GO concentration).
  2. Fig 2a: In Fig. S2 it is clearly seemed that there is a difference between the values measured with the time of acquisition, showing that 10 s/point is not good enough. I wonder why the authors presented the worse data in Fig 2a.
  3. Fig 2b: Data at lower shear rates don’t seem reliable. It can be due to the rheometer torque limit or to the time to reach the steady state regime (which is higher for low shear rates).
  4. 7, l 279: “? = ?0 + ??̇ is the shear rate” , please correct
  5. Please define what means (physically speaking) the “dynamic flow stress”. How does it compares to the yield stress? This is not clear. It is worth mentioning that the cross over point depends, for instance, in the frequency.

Reviewer 4 Report

The manuscript titled “Aging of concentrated aqueous graphene oxide suspensions 2 seen by rheology and Raman spectroscopy” investigates high concentration (10 20 mg/mL) aqueous GO suspensions over a two year time scale. In addition to steady shear tests the dynamic behavior of the suspensions was studied in more detail by transient shear and frequency sweep measurements. Both the viscosity and the dynamic moduli increased with age, particularly  within the first year. The results of the complementary Raman spectroscopic studies indicate that  the change in the rheological behavior with aging results from a slow oxidation process occurring  in the highly acidic aqueous medium during the relatively long-term storage.

There are a lot of manuscripts about the aging of graphene oxide for this reason the manuscript poor of novelty could be reconsider for publication after a major revision.

Firstly, the author must improve the introduction and discussion to clarify the novelty of results.

The authors must be improve the morphological characterization. They must improve the quality of TEM micrographs (figure 1),  add the scale bar, and improve the clarity of caption. They must be improve the quality of Figure 10 and discussion.

Furthermore, to improve the discussion about the Raman results they could  show the XRD results or better to add the XPS or FTIR characterizations.

The list of reference must be verified.

Round 2

Reviewer 3 Report

I agree with the authors answers and have no further comments.

Reviewer 4 Report

Following the suggests of reviewer, the manuscript have been modified and for this can be published.